# Synthesis and Biological Investigation of Bile Acid-Paclitaxel Hybrids

**DOI:** 10.3390/molecules27020471

**Published:** 2022-01-12

**Authors:** Elisabetta Melloni, Elena Marchesi, Lorenzo Preti, Fabio Casciano, Erika Rimondi, Arianna Romani, Paola Secchiero, Maria Luisa Navacchia, Daniela Perrone

**Affiliations:** 1Department of Translational Medicine and LTTA Centre, University of Ferrara, 44121 Ferrara, Italy; elisabetta.melloni@unife.it (E.M.); fabio.casciano@unife.it (F.C.); erika.rimondi@unife.it (E.R.); arianna.romani@unife.it (A.R.); paola.secchiero@unife.it (P.S.); 2Department of Environmental and Prevention Sciences, University of Ferrara, 44121 Ferrara, Italy; mrclne@unife.it; 3Department of Chemical and Pharmaceutical Sciences, University of Ferrara, 44121 Ferrara, Italy; prtlnz@unife.it; 4Interdepartmental Research Center for the Study of Multiple Sclerosis and Inflammatory and Degenerative Diseases of the Nervous System, University of Ferrara, 44121 Ferrara, Italy; 5Institute of Organic Synthesis and Photoreactivity, Italian National Research Council, 40129 Bologna, Italy

**Keywords:** bile acids, paclitaxel, ursodeoxycholic acid, chenodeoxycholic acid conjugation, pacific blue, hybrids drugs, anticancer activity

## Abstract

Chenodeoxycholic acid and ursodeoxycholic acid (CDCA and UDCA, respectively) have been conjugated with paclitaxel (PTX) anticancer drugs through a high-yield condensation reaction. Bile acid-PTX hybrids (BA-PTX) have been investigated for their pro-apoptotic activity towards a selection of cancer cell lines as well as healthy fibroblast cells. Chenodeoxycholic-PTX hybrid (CDC-PTX) displayed cytotoxicity and cytoselectivity similar to PTX, whereas ursodeoxycholic-PTX hybrid (UDC-PTX) displayed some anticancer activity only towards HCT116 colon carcinoma cells. Pacific Blue (PB) conjugated derivatives of CDC-PTX and UDC-PTX (CDC-PTX-PB and UDC-PTX-PB, respectively) were also prepared via a multistep synthesis for evaluating their ability to enter tumor cells. CDC-PTX-PB and UDC-PTX-PB flow cytometry clearly showed that both CDCA and UDCA conjugation to PTX improved its incoming into HCT116 cells, allowing the derivatives to enter the cells up to 99.9%, respect to 35% in the case of PTX. Mean fluorescence intensity analysis of cell populations treated with CDC-PTX-PB and UDC-PTX-PB also suggested that CDC-PTX-PB could have a greater ability to pass the plasmatic membrane than UDC-PTX-PB. Both hybrids showed significant lower toxicity with respect to PTX on the NIH-3T3 cell line.

## 1. Introduction

Paclitaxel (PTX) is a natural-based cancer drug, originally derived from the bark of the Pacific yew tree, proven to be effective in treating numerous cancer types. PTX anticancer activity is associated with its capacity in promoting microtubule assembly leading to mitotic arrest [1,2,3,4,5]. Approved by the Food and Drug Administration in 1992, PTX is among the most affordable and best-selling chemotherapy drug, with annual sales of over $1 billion. However, to overcome some limitations towards a wider framework of clinical application, many studies have been addressed to obtain smarter targeting PTX over the years. In the early stage of PTX’s development studies, the attention was mainly focused on its low water solubility. Therefore, diverse prodrug strategies have been developed to make PTX more efficient [6]. The very important role of taxoids in anticancer therapy, as well as their non-negligible side effects and drug resistance, have prompted numerous efforts devoted to the structure-activity relationship (SAR) study in order to highlight the key points of such complex molecules. SAR investigation has established that the C13 side chain, the ester groups at C2 and C4 and the rigid core represented by the oxetane ring (Figure 1) are actually strictly connected with taxoids biological activity [7,8,9,10,11]. This discovery has driven the approaches to new PTX derivatives. Many efforts have been devoted to the synthesis of modified PTX on the baccatin core as well as on the phenylisoserine side chain. Kingston et al. [12] reported the synthesis and biological evaluation of a series of PTX-steroids conjugated at the 2′-OH against a selection of solid tumors. In particular, all new conjugates were found active against the ovarian cancer cells A2780 even though, to a lesser extent, they were less active than PTX alone. Wittman et al. [13] reported the synthesis of PTX-chlorambucil hybrids by conjugation at the baccatin core that was successfully tested in vitro and in vivo in M109 and PTX resistant M109/taxlR cancer cells. A much less successful approach is represented by the replacing of the baccatin core with synthetic or natural products [14,15,16]. The new mimics of taxoids have failed to exceed the biological activity of parent PTX in both microtubule disassembly assays and cytotoxicity against numerous cancer cell types. 

For a long time, some of us have been dedicated to the design, synthesis and biological evaluation for anticancer activity of bile acids (BA)-based conjugates containing a natural molecule displaying intrinsic biological activity, such as a nucleoside [17,18] and dihydroartemisinin [19,20], or a molecule with photoinduced biological activity [21]. The design of hybrid molecules represents in cancer therapy an interesting approach to enhance biological activity and to reduce multidrug resistance. Hydroxyl and carboxyl groups of bile acids (Figure 1) can be exploited for the covalent linking of drugs to improve their physical-chemical and pharmacokinetic properties, thanks to the well-established amphiphilic features of BA. In light of their biological properties, as well as different chemical/biochemical features, BA can be considered an interesting platform for conjugation with pharmacophores [22]. As reported above, over the years, many attempts to improve PTX anticancer activity and selectivity through conjugation have been performed, and the research on this direction never stopped. In this light, and in continuation of our efforts to design BA-based conjugates, herein we report the conjugation of PTX to two different bile acids with the aim of exploring BA-PTX biological activity with respect to the parent PTX. Chenodeoxycholic acid (CDCA) and ursodeoxycholic acid (UDCA) were selected as PTX conjugation partners. CDCA and UDCA differ from each other in the absolute configuration at C7-OH and show different physical-chemical properties being CDCA more lipophilic than UDCA (Figure 1). 

Bile acids are endogenous small molecules able to inhibit themselves with cell proliferation on several cancer cell lines through many mechanisms, among others apoptosis, membrane alterations, modulation of nuclear receptors and oxidative stress [23]. Some reported evidence suggested that the hydrophobicity of bile acids is responsible for their cytotoxicity [24]. Furthermore, depending on the conditions, the more hydrophilic bile acids, including UDCA, can act either as anti- or as pro-apoptotic molecules [25,26]. 

However, BA’s relatively low intrinsic cytotoxicity (IC50 > 100 μM) prevented their clinical use and suggested their conjugation. On the other hand, due to their favorable toxicity profile and biocompatibility BA have received considerable interest in drug delivery research. PTX delivery and selectivity have been improved by using conventional drug carrier systems, for instance, encapsulation into nanosystems such as nanomicells [27] and solid lipid nanoparticles [28], or polymers conjugation [29]. BA, due to its amphiphilic properties, can act as an absorption enhancer by improving drugs bioavailability through the process of micellar solubilization and by acting as permeation-modifying agents [30]. Therefore, BA conjugation with drugs characterized by either poor aqueous solubility or low membrane permeability can be effective in enhancing bioavailability. Generally, the enhancement of bioavailability and oral adsorption is directly correlated to the hydrophobicity of BA. Furthermore, the chemical conjugation of drugs to BA may improve the oral bioavailability of parent compounds thanks to the ability of BA to withstand enzymatic and gastric degradations and to promote the active absorption via ileum apical sodium-dependent bile acid transporter (ASBT) [30].

In the present study, we approached new PTX hybrids by preserving the baccatin core and using the 2′-OH of phenylisoserine moiety as a conjugation point of paclitaxel (Figure 1). On the other side, two BA with different hydrophobicity, CDCA and UDCA were linked at the C24 position through a condensation reaction yielding the corresponding BA-PTX ester derivatives (Figure 1). The cytotoxicity of BA-PTX conjugated was evaluated towards two leukemic (HL60 and NB4) and two colon carcinoma (RKO and HCT116) cell lines.

To better investigate BA-PTX hybrids biological activity, also fluorescent probes were synthesized. To achieve more drug-like probes, we conjugated Pacific Blue (PB) to derivatives of both BA-PTX hybrids exploiting the hydroxyl group at position C7 of baccatin core accordingly to a modified procedure [31].

## 2. Results and Discussion

### 2.1. Synthesis of PTX Derivatives

#### 2.1.1. Synthesis of BA-PTX Hybrids

A condensation reaction between UDCA and CDCA and PTX, mediated by 1-ethyl-3-(3-dimethylaminopropyl)carbodiimide hydrochloride (EDC) as a water-soluble coupling reagent, was used to prepare hybrids UDC-PTX and CDC-PTX (Figure 1).

After chromatographic purification, the desired BA-PTX compounds were obtained in high yield.

#### 2.1.2. Synthesis of BA-PTX-PB and PTX-PB

To synthesize a fluorescent analogue of our hybrid compounds, we chose Pacific Blue as a fluorophore on the basis of several features of this coumarin dye. Essentially, as previously reported by Lee et al. [31], we assessed that this small and not very polar fluorophore could not affect the ability of hybrids to enter the cells. Additionally, Pacific Blue is a bright blue fluorescent dye optimally excited by the 405 nm line of violet laser in flow cytometry with good resistance to photobleaching. The synthesis of the fluorophore bearing hybrids BA-PTX-PB was achieved by modification of a previously reported procedure [31] (see Figure 2). Following the silylation of PTX, the fluorenylmethoxycarbonyl (Fmoc) glycine derivative **1** was obtained with improved yields when *N*,*N*′-Dicyclohexylcarbodiimmide (DCC) was employed in substitution of EDC hydrochloride as a coupling reagent. On the other hand, the resulting dicyclohexylurea side-product proved to be more difficult to remove from the reaction mixture than the water-soluble analogue EDC derived urea. The removal of the 2′-*O*-silyl group of compound **1** was performed by using a solution of hydrogen fluoride-pyridine (HF-Py) and allowed us to carry out the conjugation with the carboxylic group of unprotected UDCA and CDCA. The condensation reaction took place at room temperature within 3 h in the presence of DCC, affording esters **3** and **4** in 86 and 74% yields, respectively, after chromatographic purification. After removal of the Fmoc group from glycine in mild conditions, the amidation with Pacific Blue-*N*-hydroxysuccimide (PB-NHS) ester afforded desired fluorescent UDC- and CDC-PTX-PB hybrids in satisfactory yields. To obtain the purity level required for biological tests, Pacific Blue hybrids were purified by reverse-phase high performance liquid chromatography (HPLC).

The synthesis of PTX-PB depicted in Figure 2 was performed following the Lee et al. procedure [31]. Briefly, starting from compound **1,** amidation with Pacific Blue-NHS ester, was accomplished by the removal of the Fmoc group, and the final desilylation step was carried out by using a tetrabutylammonium fluoride (TBAF) solution.

### 2.2. Chemical Stability of BA-PTX HYBRIDS

Chemical stability of PTX, CDC-PTX and UDC-PTX at pH = 3 and 8 in MeOH/water solution was assessed by means of HPLC analyses. All compounds tested were found stable in the range of pH examined up to 96 h. Analogously, the chemical stability of PTX, CDC-PTX and UDC-PTX in cell culture medium DMEM was studied. A time course at 0, 4, and 24 h showed all compounds were stable also in the cell culture medium.

### 2.3. Biological Evaluation

#### 2.3.1. Comparative Effects of PTX and PTX Derivatives on Tumor Cell Lines

The cytotoxic activity of PTX derivatives was evaluated in leukemic (HL60 and NB4) and colon carcinoma (RKO and HCT116) cell lines. As shown in Figure 2A,B, CDC-PTX, used at the concentration of 2 μM, caused a significant reduction of cell viability on HL60 and NB4 cell lines either after 24 (HL60, 39.11 ± 3.38%; NB4, 52.44 ± 5.80%, vs. UNT set to 100%) and 48 h (HL60, 1.67 ± 0.49%; NB4, 29.26 ± 4.10%, vs. UNT set to 100%) of treatment and this effect was comparable to that of PTX (HL60, 24 h: 46.74 ± 5.53%, 48 h: 1.64 ± 0.82%; NB4, 24 h: 58.38 ± 4.07%, 48 h: 34.76 ± 4.76% vs. UNT set to 100%). Conversely, UDC-PTX derivative did not significantly affect NB4 cells viability at any of the time-points considered (24 h: 88.10 ± 10.32%, 48 h: 87.52 ± 2.58% vs. UNT set to 100%), but slightly reduced HL60 viability at both 24 and 48 h after treatment (24 h: 80.78 ± 4.95%, 48 h: 65.00 ± 8.61% vs. UNT set to 100%). These data were also confirmed by MTT assays data analysis Appendix A. PTX and CDC-PTX also induced a significant and comparable apoptotic effect on HL60 (PTX, 90.38 ± 1.62; CDC-PTX: 90.07 ± 1.57%) and NB4 leukemic cells (PTX, 42.35 ± 7.98%; CDC-PTX: 43.84 ± 8.34%) after 48 h of treatment (Figure 2C,D). However, it should be noted that also UDC-PTX showed a significant apoptotic effect only on the HL60 cell line at 48 h of treatment (36.62 ± 7.93%), but this effect was extremely lower if compared to that of PTX and CDC-PTX.

The comparative effects of PTX and its derivatives were also evaluated on RKO and HCT116 cell lines. In this case, as previously described by El-Sayed et al. [32], cells were treated for 4 h and then the medium was replaced with a fresh one. The cytotoxicity analyses were performed after further 72 h and showed that treatment with 5 μM PTX or CDC-PTX resulted in a significant reduction of cell viability in both colon-carcinoma cell lines (RKO, PTX: 3.57 ± 1.11%, CDC-PTX: 3.35 ± 0.48%; HCT116, PTX: 6.02 ± 1.74%, CDC-PTX: 7.21 ± 1.71% vs. UNT, set to 100%), as shown in Figure 3. Accordingly, PTX and CDC-PTX induced a significant increase in the apoptotic rate in RKO and HCT116 cells, even if PTX seemed to be more effective than CDC-PTX in inducing cell death (RKO, PTX: 88.54 ± 2.29%, CDC-PTX: 79.78 ± 4.04%; HCT116, PTX: 74.91 ± 3.70%, CDC-PTX: 66.71 ± 2.88%).

On the other hand, UDC-PTX derivative did not show any significant apoptotic effect on RKO, but it seemed to be significantly effective in causing apoptosis in HCT116 (37.88 ± 5.23%) and in inducing a reduction of cell viability on both cell lines (RKO: 65.52 ± 13.82%; HCT116: 35.25 ± 7.09%, vs. UNT set to 100%), even if much less than PTX, as indicated by the significance of the viability and apoptosis data of PTX vs. UDC-PTX.

Of note, neither CDCA nor UDCA were effective in inducing cytotoxicity both on leukemic and colon carcinoma cell lines (Figure 2, Figure 3 and Appendix A).

The data described for HCT116 cell lines were also supported by the xCELLigence analyses of cell viability/proliferation, performed using 5 μM of PTX and of its derivatives. As shown in Figure 4, treatment with PTX and CDC-PTX evidenced a low significant difference between these two treatments in reducing the Cell Index with respect to untreated cells (set to 1) only at 24 h, but no significant differences resulted at 48 and 72 h. Moreover, the effect induced by UDC-PTX, although significant with respect to the untreated cells, was nonetheless lower compared to those of PTX and CDC-PTX, further confirming the data obtained by viability and apoptosis evaluation.

#### 2.3.2. The CDC-PTX Derivative Showed Lower Toxicity with Respect to PTX on NIH-3T3 Cell Line

In the next series of experiments, the cytotoxic effect of PTX and its derivatives was assessed on NIH-3T3 fibroblast cell line, used as a healthy control for the study. As shown in Appendix A, 5 μM PTX significantly reduced cell viability after 4 h of treatment and 72 further hours of incubation in fresh medium (9.39 ± 1.94%, vs. UNT set to 100%). CDC-PTX and UDC-PTX, used at the same concentration and experimental conditions, caused a low cell viability reduction (CDC-PTX: 76.71 ± 4.11%, UDC-PTX: 72.50 ± 3.68% vs. UNT set to 100%), however not comparable to that induced by PTX. Moreover, in line with the cell viability results, PTX induced a potent apoptotic signal that led to a significant increase in the apoptotic rate (70.53 ± 4.26%), whereas the treatment with BA-PTX derivates did not show any significant induction of apoptosis (CDC-PTX: 18.23 ± 3.72%, UDC-PTX: 20.83 ± 3.70%). Even in this case, CDCA and UDCA did not significantly impair cell viability and apoptotic rate.

#### 2.3.3. Condensation of Bile Acids to PTX Ameliorated Its Incoming into the Tumor Cells

In the last series of experiments, HCT116 cells were treated with the Pacific Blue bearing compounds, in parallel to the non-conjugated ones, to verify potential differences in the ability of the different hybrids to pass through the plasmatic membrane. After treatment with CDC-PTX-PB and UDC-PTX-PB, the percentage of PB positive cells reached 99.92% ± 0.02 and 99.88% ± 0.04 respectively, while PTX-PB treatment caused the staining of only 34.69% ± 6.50 of the cells, as clearly shown in Figure 5. These data indicated that the condensation of bile acids to PTX significantly ameliorated its incoming into HCT116 cells, allowing the derivatives to enter the totality of the cells. Moreover, the significantly different fluorescence intensity of cells treated with CDC-PTX-PB with respect to UDC-PTX-PB, as indicated from the MFI (median fluorescence intensity) of these two cell populations (28125 ± 5372 for CDC-PTX-PB vs. 5372 ± 452 for UDC-PTX-PB, *p* < 0.001), suggested that CDC-PTX-PB could have a greater ability to pass the plasmatic membrane if compared to UDC-PTX-PB. As expected, cells treated with the non-bearing compounds did not show any fluorescence signal.

In summary, two novel BA-PTX hybrids based on CDCA and UDCA have been synthesized and tested for their anticancer activity towards a selection of leukemia and colon cancer cells. Whereas CDC-PTX displayed similar anticancer activity to PTX towards all cancer cell lines tested, UDC-PTX was found much less cytotoxic than CDC-PTX and PTX, exhibiting some anticancer activity only towards HCT116 cancer cells. Moreover, both BA-PTX hybrids showed significantly lower cytotoxicity towards NIH-3T3 non-tumoral cells with respect to unconjugated PTX. BA-PTX hybrids ability to pass through the plasmatic membrane has also been tested by using the newly synthesized fluorescent analogue BA-PTX-PB hybrid compounds as well as PTX-PB for a direct comparison. Interestingly, in the case of HCT116 cancer cells, both BA-PTX hybrids showed a higher ability to enter the cancer cells with PB positive cells up to 99.9% with respect to PTX-PB with PB positive cells up to only 35%.

The remarkable different cytotoxicity displayed by CDC-PTX and UDC-PTX hybrids clearly indicate that the nature of the bile acid conjugated can influence PTX anticancer activity. The different ability to enter cancer cells also proves that the presence of a BA unit can influence some physical-chemical properties, as also demonstrated for other bile acids conjugated drugs [18,19,20,33]. The greater ability to pass the plasmatic membrane found for CDC hybrid with respect to the UDC one can, at least in part, account for CDC-PTX higher anticancer activity with respect to UDC-PTX. In addition, hydrophobic BA including CDCA are known to be somehow more cytotoxic than hydrophilic ones such as UDCA.

Overall, the data herein presented, even though very preliminary, paved the way to further studies directed to tuning PTX anticancer activity through bile acid conjugation. Indeed, chemical conjugation with BA could be considered to investigate oral forms of taxoids thanks to some beneficial features of bile acids, such as their ability to target bile acid transporters and to localize in specific organs.

## 3. Materials and Methods

Commercial Paclitaxel (PTX) was purchased by Carbosynth (Compton, Berkshire, UK). Pacific blue activated as *N*-Hydroxysuccinimide ester (PB-NHS) was purchased from AAT Bioquest (Sunnyvale, CA, USA). UDCA and CDCA, were kindly furnished by ICE SpA (Reggio Emilia, Italy). All the chemicals were used without further purification. The reactions were monitored by thin layer chromatography (TLC) on pre-coated silica gel F_254_ plates (thickness 0.25 mm, Merck, Darmstadt, Germany). UV light spotted the presence of PTX and Pacific Blue, while phosphomolybdic acid solution was used as a spray to develop steroids spots. Flash column chromatography was performed on silica gel (60 a, 230–400 mesh). NMR spectra were recorded with a Varian Mercury 400 MHz instrument (Varian, Palo Alto, CA, USA), or Varian Mercury 300 MHz instrument (Varian, Palo Alto, CA, USA). HRMS spectra were acquired with a Waters Micromass ZQ instrument (Waters Corp, Milford, MA, USA). Preparative and analytical HPLC was executed on Xterra C18 column (Waters Corp., Milford, MA, USA) with a Jasco LC-2000 plus instrument (Jasco, Easton, MD, USA). Tetrahydrofuran (THF) was used as freshly distilled on sodium/benzophenone with standard procedures. *N*,*N*-dimethylformamide (DMF) was dried over freshly activated molecular sieves for at least one day before use.

### 3.1. General Procedure for the Synthesis of BA-PTX Esters

In a closed cap vial, a solution of 0.058 mmol (50 mg) of PTX and 0.064 mmol of the required bile acid in 3.5 mL of anhydrous DMF was prepared. The solution was stirred in an ice bath, and 0.064 mmol of 4-dimethylaminopyridine (DMAP) and EDC were added, then allowed to warm to room temperature and stirred overnight. The reaction mixture was then diluted with 15 mL of diethyl ether and poured into a separatory funnel. The organic phase was extracted three times with an equal volume of water in order to remove DMF and DMAP. The diethyl ether solution was dried by adding anhydrous sodium sulfate ad evaporated under vacuum. The resulting white solid was purified by flash chromatography to obtain the desired ester.

#### 3.1.1. UDC-PTX

EtOAc/Cyclohexane 4:1; white amorphous solid, 86% yield.

^1^H-NMR (400 MHz, CDCl_3_)selected data: δ 8.15–8.11 (m, 2H), 7.77–7.70 (m, 2H), 7.61–7.30 (m, 11H), 6.88 (d, *J* = 9.2 Hz, 1H), 6.29 (s, 1H), 6.27–6.21 (m, 1H), 5.94 (dd, *J* = 9.2, 3.2 Hz, 1H), 5.68 (d, *J* = 7.2 Hz, 1H), 5.51 (d, *J* = 3.4 Hz, 1H), 4.97 (d, *J* = 7.6 Hz, 1H), 4.44 (dd, *J* = 10.9, 6.7 Hz, 1H), 4.31 (d, *J* = 8.3 Hz, 1H), 4.20 (d, *J* = 8.0 Hz, 1H), 3.81 (d, *J* = 7.0 Hz, 1H), 3.61–3.53 (m, *J* = 10.3, 5.4 Hz, 2H), 2.52 (dd, *J* = 21.7, 12.1 Hz, 2H), 2.44 (s, 3H), 2.40–2.27 (m, 3H), 2.23 (s, 3H), 1.94 (d, *J* = 1.2 Hz, 3H), 1.68 (s, 3H), 1.23 (s, 3H), 1.13 (s, 3H), 0.94 (s, 3H), 0.87 (d, *J* = 6.3 Hz, 3H), 0.64 (s, 3H).

^13^C-NMR (101 MHz, CDCl_3_) selected data: δ 203.78, 171.23, 169.76, 168.07, 167.02, 166.99, 142.77, 137.01, 133.65, 132.20, 132.10, 132.01, 130.20, 129.03, 128.73, 128.43, 127.06, 126.52, 84.43, 81.03, 79.14, 76.43, 75.57, 75.07, 73.71, 72.09, 71.71, 71.40, 71.32, 58.49, 55.68, 54.80, 52.83, 45.56, 43.73, 43.16, 42.39, 40.11, 39.14, 37.22, 36.87, 35.55, 35.07, 34.89, 34.04, 30.73, 30.29, 28.59, 26.83, 23.35, 22.67, 22.10, 21.14, 20.83, 18.25, 14.83, 12.12, 9.59.

HRMS (ESI, *m*/*z*) [M + H]^+^ calculated for [C_71_H_90_NO_17_]^+^ 1129.2887; found 1129.6195.

#### 3.1.2. CDC-PTX

EtOAc/Cyclohexane 2:1 to 4:1; white amorphous solid, 94% yield. 

^1^H-NMR (400 MHz, CDCl_3_) selected data: δ 8.17–8.09 (m, 2H), 7.74 (d, *J* = 7.2 Hz, 2H), 7.68–7.29 (m, 11H), 6.89 (d, *J* = 9.1 Hz, 1H), 6.30 (s, 1H), 6.28–6.17 (m, *J* = 8.3 Hz, 1H), 5.94 (dd, *J* = 9.2, 3.1 Hz, 1H), 5.68 (d, *J* = 7.1 Hz, 1H), 5.51 (d, *J* = 3.3 Hz, 1H), 4.97 (d, *J* = 7.8 Hz, 1H), 4.44 (dd, *J* = 10.8, 6.5 Hz, 1H), 4.31 (d, *J* = 8.3 Hz, 1H), 4.20 (d, *J* = 8.4 Hz, 1H), 3.86–3.77 (m, 2H), 3.51–3.39 (m, 1H), 2.56 (s, 1H), 2.45 (s, 3H), 2.41–2.25 (m, 3H), 2.24 (s, 3H), 1.94 (s, 3H), 1.68 (s, 3H), 1.23 (s, 3H), 1.13 (s, 3H) 0.90 (s, 3H), 0.87 (d, *J* = 6.3 Hz, 3H), 0.62 (s, 3H).

^13^C-NMR (101 MHz, CDCl_3_) selected data: δ 203.78, 173.17, 171.27, 169.78, 168.08, 167.03, 166.98, 137.03, 133.66, 132.73, 132.21, 132.00, 130.20, 129.15, 129.02, 128.72, 128.42, 127.06, 126.54, 84.43, 81.03, 79.17, 76.42, 75.58, 75.07, 73.70, 72.07, 71.96, 71.70, 68.47, 58.50, 55.56, 52.82, 50.45, 45.56, 43.14, 42.64, 41.40, 39.83, 39.60, 39.36, 35.54, 35.28, 35.10, 35.01, 34.64, 32.82, 30.61, 30.52, 28.11, 26.80, 23.64, 22.74, 22.67, 20.82, 20.54, 18.12, 14.83, 11.75, 9.59.

HRMS (ESI, *m*/*z*) [M + H]^+^ calculated for [C_71_H_90_NO_17_]^+^ 1129.2887; found 1129.6198.

### 3.2. Synthesis of TBDMSiO-PTX-Gly-Fmoc 1

A solution of 0.682 mmol (66 mg) of TBDMSiO-PTX [24], 0.273 mmol (81 mg) of Gly-Fmoc and 0.034 mmol (4 mg) of DMAP in anhydrous DMF (4 mL) was prepared in a closed cap vial. To this solution, 0.273 (56 mg) mmol of DCC were added, and the mixture was stirred at 35 °C. After two hours, the mixture was diluted with ethyl acetate (50 mL) and washed with saturated NH_4_Cl (25 mL × 2), NaHCO_3_ (25 mL × 3) and brine (25 mL). The organic phase was then dried by adding anhydrous sodium sulfate, filtered, and evaporated under vacuum. The resulting crude compound was purified by flash chromatography (EtOAc/cyclohexane 1:1,8) to give 78 mg of pure compound **1** in 90% yield as an amorphous solid.

^1^H-NMR (400 MHz, CDCl_3_) selected data. δ 8.13 (d, *J* = 7.3 Hz, 2H), 7.82–7.71 (m, 4H), 7.71–7.58 (m, 2H), 7.56–7.27 (m, 16H), 7.09 (d, *J* = 8.9 Hz, 1H), 6.27 (t, *J* = 8.8 Hz, 1H), 6.21 (s, 1H), 5.77–5.68 (m, 3H), 5.52 (dd, *J* = 7.1, 4.8 Hz, 1H), 4.99 (d, *J* = 8.6 Hz, 1H), 4.69 (d, *J* = 1.7 Hz, 1H), 4.48–4.35 (m, 3H), 4.30–4.24 (m, 1H), 4.21 (d, *J* = 8.5 Hz, 1H), 4.15–4.05 (m, 1H), 3.97 (d, *J* = 6.9 Hz, 1H), 3.89–3.79 (m, 1H), 2.66–2.51 (m, 4H), 2.49–2.35 (dd, *J* = 15.2, 9.4 Hz, 1H), 2.27–2.08 (m, 4H), 1.99 (s, 3H), 1.96–1.86 (m, 1H), 1.83 (s, 3H), 1.24 (s, 3H), 1.17 (s, 3H), 0.81 (s, 9H), −0.02 (s, 3H), −0.29 (s, 3H).

^13^C-NMR (101 MHz, CDCl_3_) selected data. δ 201.82, 171.47, 169.92, 169.87, 169.65, 167.00, 166.88, 156.71, 144.06, 143.89, 141.26 (2 C), 138.20, 134.08, 133.76, 132.32, 131.80, 130.20, 129.01, 128.78, 128.75, 127.98, 127.71, 127.63, 127.03, 126.98, 126.38, 125.31, 125.22, 125.14, 119.97, 119.91, 83.89, 80.92, 78.60, 76.33, 75.69, 75.08, 74.39, 72.01, 71.29, 67.24, 56.10, 55.65, 47.14, 46.90, 43.34, 43.10, 35.58, 33.27, 32.59, 30.71, 26.37, 26.18, 25.52, 25.21, 24.68, 22.99, 21.38, 20.91, 18.13, 14.67, 10.87, −5.19, −5.80.

HRMS (ESI, *m*/*z*) [M + H]^+^ calculated for [C_70_H_79_N_2_O_17_Si]^+^ found 1248.1549 (M + H^+^) calculated 1248.5182

### 3.3. Synthesis of PTX-Gly-Fmoc 2

In a polypropylene vial, a solution of 0.41 mmol of Si-PTX-Gly-Fmoc 1 in anhydrous THF (7.8 mL) was prepared and stirred under nitrogen atmosphere at 0 °C. To this solution, 2.6 mL of HF-Pyridine were added slowly with the aid of a (polypropylene) syringe. (FAVS, Bologna, Italy) The mixture was allowed to reach room temperature, stirred for five hours, then diluted with EtOAc and quenched with NaHCO_3_ aqueous saturated solution. The organic phase was transferred to a separatory funnel and extracted with water and brine. The organic phase was dried with sodium sulfate and the solvent evaporated under vacuum. The resulting crude compound was purified by column chromatography (EtOAc/cyclohexane 1:1) to yield the product in 51% yield as a white powder.

^1^H-NMR (400 MHz, CDCl_3_) selected data. δ 8.17–8.03 (m, 2H), 7.86–7.71 (m, 4H), 7.69–7.28 (m, 17H), 7.10 (d, *J* = 9.0 Hz, 1H), 6.23–6.03 (m, 2H), 5.80 (dd, *J* = 8.9, 2.4 Hz, 1H), 5.66 (m, 2H), 5.54–5.42 (m, 1H), 4.94 (d, *J* = 8.1 Hz, 1H), 4.80 (s, 1H), 4.38 (d, *J* = 6.8 Hz, 2H), 4.31 (d, *J* = 8.5 Hz, 1H), 4.27 (t, *J* = 7.2 Hz, 1H), 4.18 (d, *J* = 8.4 Hz, 1H), 4.08 (dd, *J* = 17.9, 7.2 Hz, 1H), 3.92 (d, *J* = 6.9 Hz, 1H), 3.85 (dd, *J* = 17.8, 4.8 Hz, 1H), 3.71 (s, 1H), 2.66–2.50 (m, 1H), 2.38 (s, 1H), 2.36–2.29 (m, 3H), 2.20 (s, 1H), 1.91 (d, *J* = 12.4 Hz, 1H), 1.86 (s, 1H), 1.83 (s, 3H), 1.82 (s, 1H), 1.73 (s, 1H), 1.22 (s, 3H), 1.16 (s, 3H).

^13^C-NMR (101 MHz, CDCl_3_) δ 201.82, 172.60, 170.47, 169.97, 169.67, 167.11, 166.95, 156.80, 144.14, 143.98, 141.36, 140.79, 138.09, 133.91, 133.75, 132.72, 132.03, 130.26, 129.08, 128.84, 128.79, 128.42, 127.76, 127.15, 125.39, 125.32, 120.03, 83.90, 81.06, 78.57, 76.47, 75.82, 74.31, 73.29, 72.17, 67.34, 56.30, 55.03, 47.24, 47.09, 43.32, 43.18, 35.66, 33.38, 26.62, 22.63, 21.04, 20.93, 14.76, 10.91.

HRMS (ESI, *m*/*z*) [M + H]^+^ calculated for [C_64_H_65_N_2_O_17_]^+^ 1133.4239; found 1133.8087.

### 3.4. General Procedure for the Preparation of UDC-PTX-Gly-Fmoc 3 and CDC-PTX-Gly.Fmoc 4

To a solution of 0.39 mmol of the selected bile acid, 0.097 mmol of PTX-Gly-Fmoc 2 and 0.049 mmol of DMAP in 1 mL of anhydrous DMF in a closed cap vial, 0.39 mmol of DCC were added, and the mixture was stirred at room temperature. After 20 min a precipitate formed, and the completeness of the reaction was assessed by TLC after 1.5 h. The mixture was filtered through Celite and diluted with EtOAc (50 mL). The organic phase was extracted with NH_4_Cl (10 mL × 3), NaHCO_3_ (10 mL × 2), water (10 mL), and dried with anhydrous sodium sulfate. The solvent was removed under vacuum and the solid purified by column chromatography.

#### 3.4.1. UDC-PTX-Gly-Fmoc 3

EtOAc/cyclohexane 3:1; white powder, 82% yield.

^1^H-NMR (300 MHz, CDCl_3_) selected data. δ 8.21–8.03 (m, 2H), 7.82–7.71 (m, 3H), 7.70–7.29 (m, 18H), 6.98–6.77 (m, 1H), 6.27–6.13 (m, 2H), 6.00–5.86 (m, 1H), 5.71–5.64 (m, 1H), 5.57–5.45 (m, 2H), 5.04–4.88 (m, 1H), 4.48–4.33 (m, 1H), 4.33–4.21 (m, 1H), 4.21–4.12 (m, 1H), 4.09–3.98 (m, 1H), 3.99–3.91 (m, 1H), 3.89–3.74 (m, 1H), 3.71–3.62 (m, 1H), 3.55–3.31 (m, 1H), 2.52–2.40 (m, 1H), 2.37–2.24 (m, 1H), 2.19 (s, 1H), 2.04 (s, 1H), 1.99–1.95 (m, 1H).

^13^C-NMR (101 MHz, CDCl_3_) δ 202.26, 173.45, 170.18, 169.97, 169.81, 168.60, 167.29, 157.07, 144.43, 144.28, 141.92, 141.60, 137.39, 134.12, 132.48, 132.35, 130.54, 129.38, 129.09, 128.78, 127.99, 127.41, 126.96, 125.71, 125.62, 120.25, 84.23, 81.14, 79.05, 76.06, 74.76, 73.90, 72.44, 71.92, 68.80, 67.61, 56.43, 55.88, 50.75, 50.45, 47.49, 47.29, 43.59, 43.46, 43.01, 41.81, 40.22, 39.93, 39.74, 35.67, 35.38, 33.55, 33.15, 30.97, 30.83, 28.42, 26.78, 25.65, 24.94, 24.01, 23.10, 22.91, 21.55, 21.26, 20.90, 18.50, 14.97, 12.12, 11.21.

HRMS (ESI, *m*/*z*) [M + H]^+^ calculated for [C_88_H_103_N_2_O_20_]^+^ 1508.7093, found 1508.5138.

#### 3.4.2. CDC-PTX-Gly-Fmoc 4

EtOAc/cyclohexane 3:1; 74% yield white powder.

^1^H-NMR (300 MHz, CDCl_3_) selected data. δ 8.12 (d, *J* = 7.3 Hz, 2H), 7.81–7.71 (m, 3H), 7.71–7.30 (m, 18H), 7.05–6.91 (m, 1H), 6.34–6.12 (m, 2H), 6.01–5.90 (m, 1H), 5.70 (s, 1H), 5.56 (s, 1H), 4.94 (d, 1H), 4.37 (s, 2H), 4.33–4.23 (m, 1H), 4.22–4.15 (m, 1H), 4.13–4.02 (m, 1H), 3.98–3.86 (m, 1H), 3.85–3.77 (m, 1H), 3.55–3.36 (m, 1H), 2.68–2.46 (m, 2H), 2.42 (s, 1H), 2.37–2.24 (m, 1H), 2.19 (s, 1H), 0.93–0.80 (m, 1H), 0.62 (s, 3H).

^13^C-NMR (75 MHz, CDCl_3_) selected data δ 202.26, 173.45, 170.18, 169.97, 169.81, 168.60, 167.29, 157.07, 144.43, 144.28, 141.92, 141.60, 137.39, 134.12, 132.48, 132.35, 130.54, 129.38, 129.09, 128.78, 127.99, 127.41, 126.96, 125.71, 125.62, 120.25, 84.23, 81.14, 79.05, 76.06, 74.76, 73.90, 72.44, 71.92, 68.80, 67.61, 56.43, 55.88, 50.75, 50.45, 47.49, 47.29, 43.59, 43.46, 43.01, 41.81, 40.22, 39.93, 39.74, 35.67, 35.38, 33.55, 33.15, 30.97, 30.83, 28.42, 26.78, 25.65, 24.94, 24.01, 23.10, 22.91, 21.55, 21.26, 20.90, 18.50, 14.97, 12.12, 11.21.

HRMS (ESI, *m*/*z*) [M + H]^+^ calculated for [C_88_H_102_N_2_O_20_]^+^ 1509.7127, found 1509.7878.

### 3.5. General Procedure for the Preparation of UDC-PTX-Gly-PB and CDC-PTX-Gly-PB

8.7 μmol of the Fmoc protected glycine conjugate previously obtained were dissolved in a solution of 20% piperidine in DMF (2 mL) and stirred at room temperature for 10 min. The piperidine was completely removed under vacuum at 40 °C and to the remaining solution in DMF (approx. 0.1 mL) 3.97 µL (0.023 mmol) of DIPEA, and 2.5 mg (7.4 μmol) of PB-NHS dissolved in 0.5 mL of anhydrous DMF were added. The mixture was stirred overnight, and the solvent completely removed under vacuum. The resulting bright yellow solid was first purified by flash chromatography, then by reverse phase HPLC: A: deionized water, B: ACN + 0.1% TFA. t = 0 min A:B 60/40 t = 20 min A:B 0/100 T = 25 min A:B 0/100 t = 26 min A:B 60/40. 1.4 mL/min flow, r.t. λ = 260 nm.

#### 3.5.1. UDC-PTX-PB

Flash chromatography: DCM/MeOH 8:2; yield 63%.

^1^H-NMR (400 MHz, CDCl_3_) selected data. δ 9.15 (dd, *J* = 7.2, 4.7 Hz, 1H), 8.76 (d, *J* = 1.3 Hz, 1H), 8.15–8.00 (m, 2H), 7.80–7.70 (m, 2H), 7.66–7.29 (m, 10H), 7.19 (dd, *J* = 9.4, 2.0 Hz, 1H), 6.97 (d, *J* = 8.8 Hz, 1H), 6.24–6.12 (m, 2H), 5.87 (dd, *J* = 8.8, 3.5 Hz, 1H), 5.74 (dd, *J* = 10.7, 7.2 Hz, 1H), 5.66 (d, *J* = 7.0 Hz, 1H), 5.47 (d, *J* = 3.6 Hz, 1H), 4.96 (d, *J* = 8.5 Hz, 1H), 4.42 (dd, *J* = 17.9, 8.0 Hz, 1H), 4.32 (d, *J* = 8.3 Hz, 1H), 4.16 (d, *J* = 8.4 Hz, 1H), 3.97 (dd, *J* = 17.4, 4.5 Hz, 1H), 3.91 (d, *J* = 6.7 Hz, 1H), 3.73–3.49 (m, 2H), 2.62–2.42 (m, 4H), 2.39 (s, 3H), 2.35–2.23 (m, 4H), 2.22 (s, 4H), 2.20–2.09 (m, 4H), 2.04 (s, 3H), 1.95 (s, 3H), 1.78 (s, 3H), 1.21 (s, 3H), 1.14 (s, 3H), 0.94 (s, 3H), 0.86 (d, *J* = 6.2 Hz, 3H), 0.62 (s, 3H).

^13^C-NMR (101 MHz, CDCl_3_) selected data δ 201.87, 173.08, 169.67, 169.39, 169.01, 168.46, 167.23, 166.87, 162.16, 159.66, 147.81, 141.74, 137.04, 133.69, 132.11, 132.01, 130.12, 128.98, 128.73, 128.43, 127.08, 126.66, 116.56, 109.98, 83.94, 80.80, 78.63, 76.45, 75.67, 74.34, 73.71, 72.28, 71.86, 71.66, 71.56, 55.93, 55.60, 55.01, 53.12, 47.18, 43.70, 43.64, 43.20, 42.39, 41.82, 40.05, 39.27, 36.92, 36.63, 35.40, 35.18, 34.82, 34.01, 30.62, 30.42, 30.04, 28.67, 26.73, 26.47, 23.37, 22.58, 21.16, 18.10, 14.73, 12.07, 10.78.

HRMS (ESI, *m*/*z*) [M + H]^+^ calculated for [C_83_H_95_F_2_N_2_O_22_]^+^ 1509.6300, found 1509.7878.

#### 3.5.2. CDC-PTX-PB

Flash chromatography: DCM/MeOH 9:1; yield 17%.

^1^H-NMR (300 MHz, CDCl_3_) selected data. δ 9.28–9.15 (m, 1H), 8.78 (d, *J* = 1.2 Hz, 1H), 8.12–8.04 (m, 2H), 7.81–7.72 (m, 2H), 7.66–7.29 (m, 10H), 7.24–7.16 (m, 1H), 7.00 (d, *J* = 9.0 Hz, 1H), 6.24–6.12 (m, 2H), 5.86 (dd, *J* = 8.9, 3.9 Hz, 1H), 5.73 (dd, *J* = 10.6, 7.0 Hz, 1H), 5.65 (d, *J* = 6.8 Hz, 1H), 5.50 (d, *J* = 3.8 Hz, 1H), 4.97 (d, *J* = 8.8 Hz, 1H), 4.48–4.36 (m, 1H), 4.32 (d, *J* = 8.5 Hz, 1H), 4.16 (m, 1H), 4.08–3.95 (m, 1H), 3.90 (d, *J* = 6.7 Hz, 1H), 3.87–3.77 (m, 1H), 3.58 (m, 1H), 2.74 (s, 1H), 2.37 (s, 1H), 1.25 (s, 3H), 1.20 (s, 3H), 1.14 (s, 3H), 0.62 (s, 3H).

HRMS (ESI, *m*/*z*) [M + H]^+^ calculated for [C_83_H_95_F_2_N_2_O_22_]^+^ 1509.6333, found 1509.7889.

### 3.6. Cell Cultures

The human leukemic cell lines HL60, obtained from the American Type Culture Collection (ATCC, Manassas, VA, USA) and NB4, purchased from DSMZ (German Collection of Microorganism and Cell Culture, Braumschweig, Germany), were cultured in RPMI-1640 medium supplemented with 10% fetal bovine serum (FBS), 2 mM l-glutamine, 100 U/mL penicillin and 100 mg/mL streptomycin (all from Gibco, Grand Island, NY, USA). RKO and HCT116 human colon-carcinoma cell lines and the mouse embryonic fibroblast NIH-3T3 cell line were obtained from ATCC and cultured in DMEM supplemented with 10% FBS, 2 mM l-glutamine, and antibiotics (all from Gibco, Grand Island, NY, USA). All the cells were maintained in an atmosphere of 5% CO_2_ and 90% relative humidity at 37 °C.

### 3.7. Cell Treatments and Evaluation of Cell Viability and Apoptosis

All PTX derivatives were reconstituted in DMSO and their effects were comparatively evaluated in HL60, NB4, RKO, HCT116 and NIH-3T3 cell lines. HL-60 and NB4 were treated for 24 and 48 h with 2 μM of each of the following compounds: PTX, CDC-PTX, UDC-PTX, CDCA and UDCA. RKO, HCT116 and NIH-3T3 cell lines were exposed to 5 μM of the same compounds for 4 h and then the treatments were removed and replaced with fresh medium for 72 h before analysis. In order to assess viability, cells were examined by Trypan blue dye exclusion and/or MTT colorimetric assay (Roche Diagnostics Corporation, Indianapolis, IN, USA) for data confirmation, following the manufacturer’s instruction. At the same time points, the percentage of apoptosis was determined by flow cytometry (FACSCalibur; BD Biosciences, San Josè, CA, USA) following Annexin V-FITC/propidium iodide (PI) double staining (Beckman Coulter Inc., Brea, CA, USA), as previously described [34]. Apoptosis data analysis was performed using the FloJo Software (Tree Star, Ashland, OR, USA).

In order to evaluate the ability of PTX derivatives to pass the plasmatic membrane, HCT116 cells were treated with 18 μM of the Pacific Blue bearing compounds (PTX-PB, CDC-PTX-PB and UDC-PTX-PB) for 4 h. After this time, treatments were removed and replaced with fresh medium for further 72 h. Pacific Blue fluorescence was then detected using the FACSAria-IIIu flow cytometer (BD Biosciences, San Josè, CA, USA)) and analyzed by using the FloJo Software (Tree Star, Ashland, OR, USA). The chosen concentration (18 μM) corresponded to the maximum quantity of DMSO in the culture medium that did not induce a vehicle-specific apoptotic effect on HCT116 cell lines analyzed at the same time points after treatment (data not shown).

### 3.8. Evaluation of Time Course Effect of PTX and PTX-BA Hybrids

The time course effects of PTX and of its derivatives on HCT116 cells were also evaluated by the xCELLigence RTCA DP Instrument (F. Hoffmann-La Roche SA, Basel, Switzerland). In particular, 3 × 10^4^ cells were seeded onto 16-well E-plates in 200 µL of complete medium and cultured in presence of 5% CO_2_ at 37 °C. After approximately 24 h from the seeding, as suggested by the manufacturer, the cells were treated with 5 μM PTX, CDC-PTX and UDC-PTX for 4 h and then the treatments were replaced with fresh complete medium. The xCELLigence RTCA DP Instrument registers in real-time the impedance values related to cell viability and proliferation and converts them in an adimensional parameter called “Cell Index” (CI). For these experiments, the impedance was measured every 15 min.

### 3.9. Statistical Analysis

All data, obtained from at least three independent experiments, were tested for normal distribution by Shapiro–Wilk normality test and for homogeneity of variance by the Brown–Forsythe test. Results were evaluated by one-way ANOVA followed by Bonferroni post hoc test (for multiple corrections) or by Mann–Whitney test (for not normally distributed data) using GraphPad Prism software, version 8.4.2 (GraphPad Software, San Diego, CA, USA). Results were expressed as mean ± standard error (SEM) of replicate experiments. Statistical significance was defined as almost *p* < 0.05.

### 3.10. Chemical Stability of PTX and PTX-BA Hybrids

HPLC analyses were performed on HPLC Dyonex Ultimate 3000 (Sunnyvale, CA, USA) equipped with a diode array UV detector. 0.5 mL samples were used as sources for the automated injection. LC-MS grade methanol was purchased from Sigma-Aldrich (St. Louis, MO, USA) in the highest available purity and was used without any further purification. Ultrapure water (resistivity 18.2 MΩ/cm at 25 °C) was produced in our laboratory by means of a Millipore Milli-Q system. The chromatographic separation was performed on a reverse-phase Zorbax C18 column (Agilent, Santa Clara, CA, USA) 4.6 × 150 mm, 5 μm, at flow rate of 1 mL/min, linear gradient H_2_O (TFA 0.5%)/MeOH from 30:70 to 5:95, detection at λ227 nm.

Chemical stability of PTX, CDC-PTX and UDC-PTX at pH = 3 and 8 in MeOH/water solution was assessed. A mother solution in methanol PTX, CDC-PTX and UDC-PTX (58, 48 and 48 μM, respectively) was prepared. In case of stability under acidic conditions, pH aqueous HCOOH 1% was added to each solution up to pH = 3; in turn, for stability under basic conditions, aqueous NaOH 0.1% was added to each solution up to pH = 8. The solutions were analyzed by HPLC at selected times (3, 24, 48, 72, 96 h). All compounds tested were found stable in the range of pH examined.

Chemical stability in cell culture medium DMEM supplemented with 10% FBS, 2 mM l-glutamine, and antibiotics: DMSO mother solutions of PTX, CDC-PTX and UDC-PTX at 20 mM concentration were prepared. Each solution was diluted in complete cell culture medium to a final concentration of 10 μM. At different times (time 0, 4 and 24 h), a 500 μL sample of each solution was taken and extracted with 500 μL of ethyl acetate. After centrifugation (5 min at 18,000 rpm), the organic phase was recovered and evaporated off. The residues were solubilized in 500 μL of mobile phase (i.e., H_2_O/MeOH 30:70) and analyzed by HPLC. The concentration of PTX, CDC-PTX and UDC-PTX during time was calculated by comparison with that of the initial solution (time 0). All compounds tested were found stable in medium up to 24 h.

## Data Availability

Not applicable.

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
