# Peer review of "Synthesis and Biological Investigation of Bile Acid-Paclitaxel Hybrids"

_molecules, 2022, doi:10.3390/molecules27020471_

Round 1
Reviewer 1 Report
It is an interesting study about synthesis and biological investigation of bile acid-paclitaxel conjugation. I suggest it for publication in Molecules after the following issues are addressed.
- It is not very clear why bile acids were designed to be conjugated to PTX.
- What is the water solubility of these new conjugates?
- The mechanism for the remarkable different cytotoxicity displayed by CDC-PTX and UDC-PTX is not clear. More discussion should be added.
- These new conjugates still needs a suitable carrier for the treatment. The authors should add some discussion about the delivery of new conjugates using nanocarriers for the better bioavailability. Several recent studies (doi.org/10.1016/j.colsurfb.2020.111524; doi.org/10.3390/pharmaceutics13070929; doi.org/10.1016/j.actbio.2020.12.024) related to PTX delivery are recommended to be included.
Author Response
It is an interesting study about synthesis and biological investigation of bile acid-paclitaxel conjugation. I suggest it for publication in Molecules after the following issues are addressed.
We thank the Reviewer for giving us the opportunity to submit a revised draft of the manuscript “Synthesis and Biological Investigation of Bile Acid-Paclitaxel Hybrids” for publication in Molecules. We have appreciated all suggestions made by the Reviewer. Please, see below, in blue, for a point-by-point response.
- It is not very clear why bile acids were designed to be conjugated to PTX.
Authors’ reply:
We thank the reviewer for this comment. The introduction has been reviewed in order to better highlight the biological relevance of bile acids either themselves and as conjugation partners
- What is the water solubility of these new conjugates?
Authors’ reply
Hybrids water solubility has not yet been investigated. In the present work they were used as PTX alone. We agree with the Reviewer that water solubility it is an important point and a deeper investigation on some chemical-physical properties of the hybrids respect to PTX will be carried out.
- The mechanism for the remarkable different cytotoxicity displayed by CDC-PTX and UDC-PTX is not clear. More discussion should be added.
Authors’ reply
We thank reviewer for this comment. In the last part of the results and discussion section (page 9, lines 131-135 of revised manuscript), we have now added some possible explanations for the very different cytotoxicity found for the two hybrids.
- These new conjugates still needs a suitable carrier for the treatment. The authors should add some discussion about the delivery of new conjugates using nanocarriers for the better bioavailability. Several recent studies (doi.org/10.1016/j.colsurfb.2020.111524; doi.org/10.3390/pharmaceutics13070929; doi.org/10.1016/j.actbio.2020.12.024) related to PTX delivery are recommended to be included.
Authors’ reply
We thank Reviewer for this comment. In the introduction section of the submitted manuscript we briefly reported about the relevance of BA in oral delivery. As suggested by the Reviewer we added some more considerations about PTX drug delivery concern highlighting also the relevance bile acids in drug delivery. The suggested papers have been included.

Reviewer 2 Report
This manuscript deals with the synthesis and bioevaluation of several compounds against a limited number of neoplastic and non-malignant cells. The work is of sufficient importance to be published but the current rendition needs substantial revision. The principal change that needs to be made is to subsume the myriad of figures into a few tables. Specifically lines 150-286 need to be placed into tables and the figures removed.
In addition the English needs to be substantially improved throughout the presentation.
Author Response
This manuscript deals with the synthesis and bioevaluation of several compounds against a limited number of neoplastic and non-malignant cells. The work is of sufficient importance to be published but the current rendition needs substantial revision.
We thank the Reviewer for giving us the opportunity to submit a revised draft of the manuscript “Synthesis and Biological Investigation of Bile Acid-Paclitaxel Hybrids” for publication in Molecules. We have appreciated all suggestions made by the Reviewer. Please, see below, in blue, for a point-by-point response.
1) The principal change that needs to be made is to subsume the myriad of figures into a few tables. Specifically lines 150-286 need to be placed into tables and the figures removed.
Authors’ reply:
We thank the Reviewer for this suggestion. Accordingly, we reduced the number of figures in the text and moved two of them in the supplementary materials to give the manuscript more fluency. On the other hand, we could not remove all the figures because they also show the statistical significances of the described data and, in the authors’ opinion, significances, and then the biological data relevance, could be not clearly understandable if reported in tables.
2) In addition, the English needs to be substantially improved throughout the presentation.
Authors’ reply: We thank the Reviewer. Accordingly, we improved the English.

Reviewer 3 Report
In this manuscript, E. Melloni et al. described the synthesis of bile acid-paclitaxel (BA-PTX) conjugates and their anticancer activity. First, authors synthesized two BA-PTX hybrids using cheno- and urso-deoxycholic acid and the experimental procedures and characterization data on all the new compounds were well described. Next, both the conjugates were tested for their anticancer activity in the leukemia and colon cancer cell lines. The α-hydroxy cholic acid conjugated paclitaxel showed similar cancer activity as paclitaxel and β-hydroxy cholic acid paclitaxel conjugate showed much less cytotoxicity than paclitaxel. In addition, authors have tested the ability of these conjugates to pass through the membrane by adding the fluorescent tether. Considering the importance and the need of more studies for cancer diseases, the publication in Molecules is recommended.
Author Response
In this manuscript, E. Melloni et al. described the synthesis of bile acid-paclitaxel (BA-PTX) conjugates and their anticancer activity. First, authors synthesized two BA-PTX hybrids using cheno- and urso-deoxycholic acid and the experimental procedures and characterization data on all the new compounds were well described. Next, both the conjugates were tested for their anticancer activity in the leukemia and colon cancer cell lines. The α-hydroxy cholic acid conjugated paclitaxel showed similar cancer activity as paclitaxel and β-hydroxy cholic acid paclitaxel conjugate showed much less cytotoxicity than paclitaxel. In addition, authors have tested the ability of these conjugates to pass through the membrane by adding the fluorescent tether. Considering the importance and the need of more studies for cancer diseases, the publication in Molecules is recommended.
We thank the Reviewer for giving us the opportunity to submit a revised draft of the manuscript “Synthesis and Biological Investigation of Bile Acid-Paclitaxel Hybrids” for publication in Molecules.
We thank the reviewer for the positive comment.

Round 2
Reviewer 2 Report
I feel the authors have addressed my concerns and the article should be published.